# Mapping Interactions Between Cytokines, Chemokines, Growth Factors, and Conventional Biomarkers in COVID-19 ICU-Patients

**DOI:** 10.3390/ijms262311419

**Published:** 2025-11-26

**Authors:** Mats B. Eriksson, Michael Marks-Hultström, Mikael Åberg, Miklós Lipcsey, Robert Frithiof, Anders O. Larsson

**Affiliations:** 1Section of Anaesthesiology and Intensive Care Medicine, Department of Surgical Sciences, Uppsala University, 751 85 Uppsala, Sweden; michael.hultstrom@mcb.uu.se (M.M.-H.); miklos.lipcsey@uu.se (M.L.); robert.frithiof@uu.se (R.F.); 2NOVA Medical School, New University of Lisbon, 1099-085 Lisbon, Portugal; 3Integrative Physiology, Department of Medical Cell Biology, Uppsala University, 751 85 Uppsala, Sweden; 4Section of Clinical Chemistry and SciLifeLab Affinity Proteomics, Department of Medical Sciences, Uppsala University, 751 85 Uppsala, Sweden; mikael.aberg@medsci.uu.se; 5Hedenstierna Laboratory, Department of Surgical Sciences, Uppsala University, 751 85 Uppsala, Sweden; 6Section of Clinical Chemistry, Department of Medical Sciences, Uppsala University, 751 85 Uppsala, Sweden; anders.larsson@akademiska.se

**Keywords:** biomarker, nucleated blood cells, COVID-19, critical illness, CRP, cytokine, ferritin, inflammation, procalcitonin

## Abstract

Severe coronavirus disease 2019 (COVID-19) is characterized by systemic hyperinflammation with cytokine and chemokine release, alongside elevations in conventional laboratory biomarkers such as C-reactive protein (CRP), ferritin, and procalcitonin (PCT). However, the interplay between cytokines, chemokines, growth factors (CCGFs), and standard biomarkers remains incompletely understood. Therefore, we aimed to evaluate associations between CCGFs and conventional biomarkers from a broad aspect, utilizing the prospective PronMed cohort of critically ill COVID-19 patients admitted to the intensive care unit (ICU) at Uppsala University Hospital. Plasma concentrations of 92 CCGFs were analyzed in each patient using the Olink Target 96 Cardiovascular II panel and analyzed in relation to conventional biomarkers and peripheral blood cell counts. Associations were evaluated using Spearman rank correlations with Benjamini–Hochberg correction for multiple testing. A total of 114 patients (median age 61 years (IQR: 19), 75% male, median SAPS-3 52 (IQR: 10) were included. Significant correlations confirmed CRP as a robust surrogate of cytokine-driven inflammation. Ferritin was strongly associated with macrophage-related markers, including IL-18, sCD163-related factors, and PARP1. PCT correlated with a wide range of CCGFs, including ADM, PGF, TRAILR2, and IL-6. Blood cell subsets also showed distinct associations with CCGFs, suggesting functional connections between cytokine signaling and hematological disturbances. Our findings demonstrate that conventional biomarkers of inflammation in COVID-19 reflect complex and distinct interaction patterns with cytokines, chemokines, and growth factors. Mapping these associations improves understanding of COVID-19 immunopathology and may inform biomarker-guided risk stratification in critical illness.

## 1. Introduction

The pandemic coronavirus disease 2019 (COVID-19), caused by severe acute respiratory syndrome coronavirus 2 (SARS-CoV-2) infections, has been characterized not only by viral pneumonia but also by systemic endothelial inflammation, acute kidney injury and myocardial manifestations [1,2,3]. Early in the pandemic, it became clear that severe cases were frequently associated with a state of hyperinflammation, in which multiple inflammatory mediators were elevated and strongly linked to adverse clinical outcomes. This prompted a plethora of questions regarding the relationships between circulating cytokines, growth factors, and chemokines, and more traditional laboratory markers of systemic inflammation such as blood cell counts, C-reactive protein (CRP), ferritin, and procalcitonin (PCT) [4,5,6].

CRP is a frequently used acute-phase reactant and was among the earliest biomarkers reported to be elevated in severe COVID-19 [7]. Multiple studies have consistently demonstrated strong associations between circulating cytokines, especially interleukin-6 (IL6), and CRP levels. The connection between CRP and IL6 is well-established, since IL6 directly induces hepatocyte CRP production. In COVID-19 cohorts, higher CRP concentrations have been tightly linked to elevated IL6 and to worse outcomes, including respiratory failure, intensive care unit (ICU) admission, and mortality [8,9].

Not only other cytokines, e.g., tumor necrosis factor-alpha, but also white blood cells have shown correlations with CRP, underscoring that CRP elevation in COVID-19 reflects a broader pro-inflammatory milieu rather than the action of IL6 alone [10]. Thus, CRP, while non-specific, remains a robust surrogate for the cytokine-driven inflammatory burden in COVID-19 [11].

Hyperferritinemia has emerged as another hallmark of severe COVID-19 [12]. Moreover, elevated ferritin and macrophage activation are mediators of thrombotic complications in COVID-19 [13]. Ferritin levels correlate not only with CRP [14] but also with biomarkers of cell damage [15].

Furthermore, ferritin correlates with sCD163, the soluble form of the CD163 receptor, which is mainly located on the surface of monocytes and macrophages, and also with interleukin-18 (IL18) [16], suggesting a mechanistic link between macrophage activation and cytokine release in COVID-19.

Procalcitonin is commonly used as an infectious biomarker, indicating disease severity, especially in the intensive care unit. Although procalcitonin is relatively specific to bacterial infections, serum procalcitonin levels correlate with disease severity, which decreases its reliability in the setting of critical illness and especially in cases of severe influenza and COVID-19 [17].

Pro-inflammatory cytokines such as IL6 and TNF-α are elevated in patients with increased PCT, and meta-analyses confirm that elevated PCT is associated with severe and fatal outcomes [18,19].

Alterations in peripheral blood cell counts, particularly decreased numbers of platelets, lymphopenia and neutrophilia, are some of the striking features of severe COVID-19 infections, and increased neutrophil-to-lymphocyte ratio has been reported to be a strong predictor of severity and mortality [20]. Associations between cytokines (IL6, IL-8, TNF-α) and blood cell abnormalities may connect cytokine signaling directly to hematological disturbances [20,21,22].

Furthermore, several chemokines, e.g., CXCL10 and CCL2, have been found at elevated levels in severe infectious disease, and their correlations with peripheral leukocyte counts point to a functional link between chemokine signaling and observed blood cell dynamics [23,24].

The inflammatory reaction consists of a myriad of interconnected reactions and events, where traditional biomarkers are involved in a complex interplay with circulating cytokines, chemokines, and growth factors (CCGFs). Still, interactions between CCGFs, several biomarkers, and circulating blood cells remain incompletely mapped. Newer technologies, such as proximity extension assays (e.g., Olink panels) enable simultaneous quantification of large numbers of CCGFs with high sensitivity and specificity.

The aim of the present study is to investigate associations between a broad panel of CCGFs and some conventional biomarkers of inflammation in patients with COVID-19 in order to expand current knowledge of COVID-19 pathophysiology and to explore whether yet-undiscovered interactions may be identified. Since severe COVID-19 represents a state of thromboinflammation, endotheliopathy and cardiovascular mortality [25], we decided to focus on a cytokine panel for biomarkers associated with cardiovascular morbidity and mortality.

## 2. Results

### 2.1. Patient Characteristics

This cohort consisted of one hundred and fourteen patients aged 24–86 years, of which eighty-six were males. The median age was 61 years and the interquartile range (IQR) was 52–71 years. Median BMI (kg/m^2^) was 28 IQR (25–33). Fifty-three percent had pre-existing hypertension and twenty-seven percent had diabetes. Median duration of COVID-19 before ICU admission was 10 days (IQR: 8–12) and SAPS-3 was 52 (IQR: 47–57). At ICU admission median C-reactive protein (CRP; mg/L) was 164 (IQR: 113–235), whereas median procalcitonin (microg/L) was 0.46 (IQR: 0.19–1.1) and median ferritin (microg/L) was 1284 (IQR: 543–2395).

### 2.2. Associations Between Inflammatory Biomarkers (CRP, PCT, IL-6, Ferritin) and Peripheral Blood Nucleated Cells

CRP was significantly correlated with both neutrophil cells and leucocytes, respectively. PCT was significantly correlated with leucocytes and IL6 was significantly correlated with neutrophil cells (Figure 1). The exact correlations and their significance levels are presented in Appendix A.

Significant associations between biomarkers and CCGFs are displayed in Table 1.

Interactions between conventional inflammatory biomarkers and CCGFs are displayed as STRING images.

### 2.3. Associations Between CRP and CCGFs

CRP exhibited 6 significant associations out of the quantified CCGFs (Figure 2). IL6 was the one that was most associated with CRP, followed by SCF and IL1Ra.

### 2.4. Associations Between PCT and CCGFs 

Procalcitonin was significantly associated with 19 CCGFs, out of which the most expressed association was noted with ADM, followed by TNFRSF10A and in turn by CTSL1 (Figure 3). CRP was significantly more associated with procalcitonin than IL6 was.

### 2.5. Associations Between IL6 and CCGFs

IL6 was most significantly associated with ADM, CRP, and CCL3 (Figure 4). PCT exhibited the second weakest association with IL6.

### 2.6. Associations Between Ferritin and CCGFs

Ferritin was significantly associated with ten CCGFs. The three most expressed associations were between ferritin and, in the following order, CTSL1, PARP1, and IL18 (Figure 5).

## 3. Discussion

In this prospective cohort of critically ill COVID-19 patients, we investigated the associations between conventional inflammatory biomarkers (CRP, PCT, ferritin, and blood cell counts) and a broad panel of circulating cytokines, chemokines, and growth factors (CCGFs) using proximity extension assays. Our data provide new insights into the complex inflammatory milieu of severe COVID-19 and highlight several previously unrecognized links between routine clinical biomarkers and mediators of cytokine signaling, macrophage activation, and endothelial dysfunction.

As expected, CRP showed a strong and highly significant association with IL6, confirming prior studies that have consistently demonstrated the IL6–CRP axis as a central pathway in COVID-19-related inflammation [26]. Beyond IL6, we identified significant associations between CRP and IL1 receptor antagonist (IL1Ra), stem cell factor (SCF), and extracellular matrix components such as decorin (DCN) and prolargin (PRELP). These findings suggest that CRP may not only reflect hepatocyte stimulation by IL6 but may also serve as a surrogate for broader cytokine-driven tissue and extracellular matrix remodeling processes during severe infection.

Procalcitonin, typically regarded as a biomarker of bacterial superinfection [27], demonstrated extensive associations with multiple CCGFs, including adrenomedullin (ADM), placental growth factor (PGF), programmed death-ligand 2 (PD-L2), and TRAIL receptor 2 (TRAILR2). Interestingly, PCT was also tightly linked to IL6 and TNF superfamily members, underscoring that elevated PCT in COVID-19 is not solely attributable to bacterial coinfection but may instead reflect a state of profound systemic inflammation. This aligns with prior meta-analyses showing that PCT is associated with adverse outcomes in COVID-19, irrespective of concurrent bacterial infection. The broad CCGF interaction profile of PCT in our study reinforces the notion that this biomarker reflects inflammatory severity rather than etiology in critical illness [28,29].

Hyperferritinemia, a hallmark of severe COVID-19 [12], also displayed strong associations with cytokines and enzymes linked to macrophage activation and oxidative stress. In particular, ferritin correlated with IL18, heme oxygenase-1 (HO-1), soluble transglutaminase 2 (TGM2), and the protease cathepsin L1 (CTSL1). These findings reinforce the role of ferritin as more than a passive bystander, instead reflecting macrophage activation and cellular stress responses that contribute to endothelial injury and thromboinflammation. The link between ferritin and PARP1 further implicates DNA damage and repair pathways in the hyperinflammatory response of COVID-19 [30].

Peripheral blood cell counts, while widely used in clinical monitoring, showed meaningful correlations with several immune mediators. For example, basophil and eosinophil counts were linked to galectin-9 (Gal-9) and receptor for advanced glycation end-products (RAGE), both molecules implicated in immune checkpoint regulation and tissue damage responses. Neutrophils and total leukocyte counts associated with ADAMTS13, GIF, and CEACAM8, highlight a functional connection between hematological changes and endothelial as well as granulocyte activity. Such findings may help explain why hematological parameters, particularly the neutrophil-to-lymphocyte ratio, consistently predict outcome in COVID-19 [20].

Our findings extend current knowledge by integrating cytokines, chemokines, growth factors, and standard biomarkers by revealing distinct and biologically coherent correlation patterns. This comprehensive mapping clarifies how conventional biomarkers reflect underlying immune pathways, which may contribute to more refined biomarker-guided risk stratification.

In this exploratory trial, we used a cardiovascular Olink panel to evaluate a broad spectrum of CCGFs reflecting cardiac injury in patients infected with SARS-CoV-2. It is a drawback that some important clinical covariates that influence inflammatory markers (e.g., immunomodulatory treatments, bacterial coinfection, comorbidities, organ dysfunction) are not evaluated regarding their potential impact on the CCGFs measured. As expected, elevated levels of IL6, CRP, ferritin, and PCT were observed, all of which have been associated with cardiovascular events [6,31,32]. Treatment with dexamethasone in severe COVID-19 may have attenuated some of the CCGFs analyzed. In a previous study, we demonstrated that IL6 was significantly associated with body mass index, high-density lipoproteins, estimated glomerular filtration rate, and triglycerides in presumed healthy individuals [33]. Thus, it cannot be excluded that certain comorbidities may influence the interpretation of IL6 levels, and possibly those of other CCGFs associated with conventional biomarkers and nucleated blood cells. Viral pneumonia has been predicted by IL6, IL27, and CRP, with distinct cytokine expression profiles differentiating viral from bacterial community-acquired pneumonia [34]. In SARS-CoV-2 infection, IL6 is one of the cytokines most clearly associated with poor clinical outcomes [35].

Thus, prospective validation studies—such as cluster analyses using artificial intelligence—may help determine whether specific cytokine patterns can guide therapeutic decision-making. If so, the Olink panel (or similar platforms) may be cost-effective, particularly when evaluated in relation to ICU care costs.

## 4. Materials and Methods

### 4.1. Study Population

This investigation was conducted as a sub-study of the single-center, prospective observational PronMed cohort, carried out at the Intensive Care Units of Uppsala University Hospital, Uppsala, Sweden, which treated critically ill COVID-19 patients during the pandemic. Adult patients with severe COVID-19 treated in intensive care were screened for eligibility. COVID-19 was confirmed by a positive polymerase chain reaction (PCR) test of a nasopharyngeal sample. Baseline characteristics, including age, sex, and BMI, were recorded at ICU admission, and comorbidity data were extracted from medical records. Data, including the Simplified Physiology Score (SAPS-3) [36,37], were collected between 14 March 2020 and 10 March 2021. Blood samples for biochemical analyses were obtained as part of routine clinical care. Plasma samples were stored at −80 °C until analyzed. The median time (IQR) from symptom onset to ICU admission was 10 days (8–12). From 22 June 2020 onward, patients with COVID-19 requiring supplemental oxygen received dexamethasone 6 mg daily. Demographic data at ICU admission have been reported previously [38].

### 4.2. Ethical Considerations

The study was conducted in accordance with the ethical principles outlined in the Declaration of Helsinki [39] and was consistent with ICH/GCP E6 (R2), the EU Clinical Trials Directive, GDPR (EU 2016/679), and applicable local regulations. Approval was obtained from the Swedish National Ethical Review Agency (Dnr 2017-043 approved 23 August 2017 by Per-Erik Nistér; amendments 2019-00169, 2020-01623, 2020-02719, 2020-05730, 2021-01469, and 2022-00526-01). Informed consent was obtained from all patients or from their next of kin if the patient was unable to provide consent. The study protocol was prospectively registered (Clinical Trials ID: NCT04316884) and conducted in accordance with relevant directives.

### 4.3. Proximity Extension Assay

The proximity extension assay (PEA) using the Olink Target 96 Cardiovascular II panel was conducted on citrate plasma samples at SciLifeLab Affinity Proteomics, Uppsala University (Olink Proteomics, Uppsala, Sweden) [40]. The PEA has been reported to provide protein plasma levels in good agreement with conventional immunoassays [41]. In brief, 1 µL of each plasma sample was incubated with pairs of oligonucleotide-labeled antibodies targeting 92 inflammation-related proteins. When both antibodies bound to the same target protein, their attached oligonucleotides hybridized and were extended by DNA polymerase to form a unique reporter sequence. These reporter molecules were then amplified by universal PCR and quantified using high-throughput real-time PCR (Fluidigm BioMark HD real-time PCR, Standard BioTools, San Francisico, CA, USA), generating Normalized Protein eXpression (NPX) values on a log2 scale. Assay performance was monitored using Olink’s internal controls (incubation, extension, and detection controls). Data processing, QC, and normalization were performed using Olink NPX Manager software v2.2.1. 311.

Analyzed cytokines, chemokines, and growth factors (CCGFs) are seen in Appendix A, where their genes, full names, and Uniprot IDs are also displayed.

### 4.4. STRING Images

Cytokine, chemokine, and growth factor interactions, together with concentration patterns, were visualized using STRING database–generated images [42]. Edge weights were calculated in STRING based on the associations identified in this study. Protein names were displayed to create interaction networks, and edge thickness reflected confidence scores, indicating the likelihood of a true interaction according to STRING’s integrated evidence. Images were exported in high resolution for figure preparation, enabling simultaneous visualization of quantitative concentration data and qualitative interaction patterns.

### 4.5. Statistical Analysis

Coefficients of variation were examined using Spearman rank correlations in Statistica (StatSoft, v14; Tulsa, OK, USA). Cytokine values below the lowest standard point were included in analyses in agreement with Olink recommendations, and no values exceeded the assay’s upper or lower standard curve limits. To account for the increased risk of false positives, due to multiple comparisons, *p*-values were adjusted using the Benjamini–Hochberg false discovery rate (FDR) method [40]. Adjusted *p*-values < 0.10, corresponding to an expected FDR ≤ 10%, were considered statistically significant. As this is an exploratory study, we used somewhat higher levels of significance to ensure the avoidance of a statistical type II (beta) error. Continuous variables are denoted as median and interquartile range [IQR (75th percentile–25th percentile)].

## 5. Conclusions

In conclusion, our findings demonstrate that conventional inflammatory biomarkers such as CRP, PCT, and ferritin reflect distinct yet overlapping aspects of the cytokine, chemokine, and growth factor network in severe COVID-19. CRP closely mirrored the IL6–driven acute-phase response, PCT revealed broader links to pro-inflammatory and endothelial mediators, and ferritin highlighted macrophage activation and oxidative stress pathways. Hematological parameters also proved to be integrated within this network. These results may turn out to have clinical implications by clarifying which routinely measured biomarkers best reflect specific immune pathways, improve risk stratification, and guide monitoring strategies in critical care. Together, these results emphasize that routine laboratory markers remain valuable as accessible surrogates of complex immune responses, and they provide new biological insights that may inform biomarker-guided monitoring and targeted therapeutic strategies in COVID-19.

## Figures and Tables

**Figure 1 ijms-26-11419-f001:**
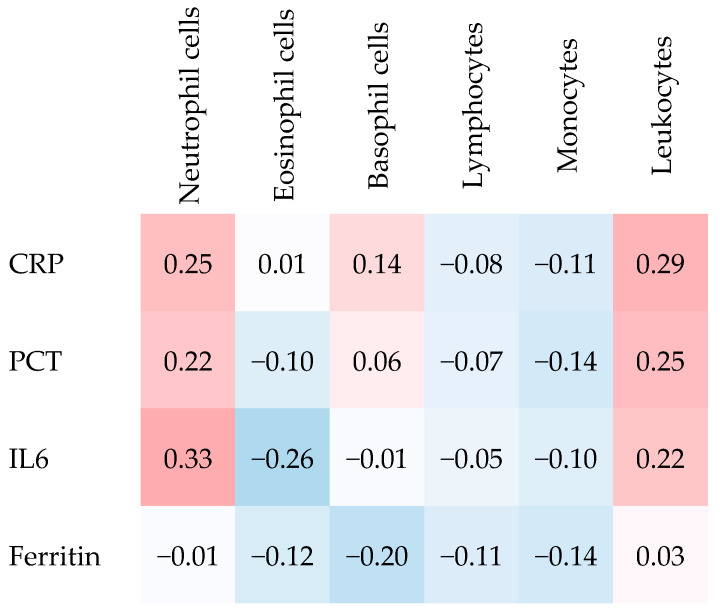
Heatmap displaying Spearman’s rank correlation coefficients. CRP = C-reactive protein; PCT = Procalcitonin; IL6 = Interleukin-6. Increasingly warmer colors (light red to darker red) indicate stronger positive correlations, whereas cooler colors (light blue to blue) indicate more expressed negative correlations. No color denotes absence of correlation.

**Figure 2 ijms-26-11419-f002:**
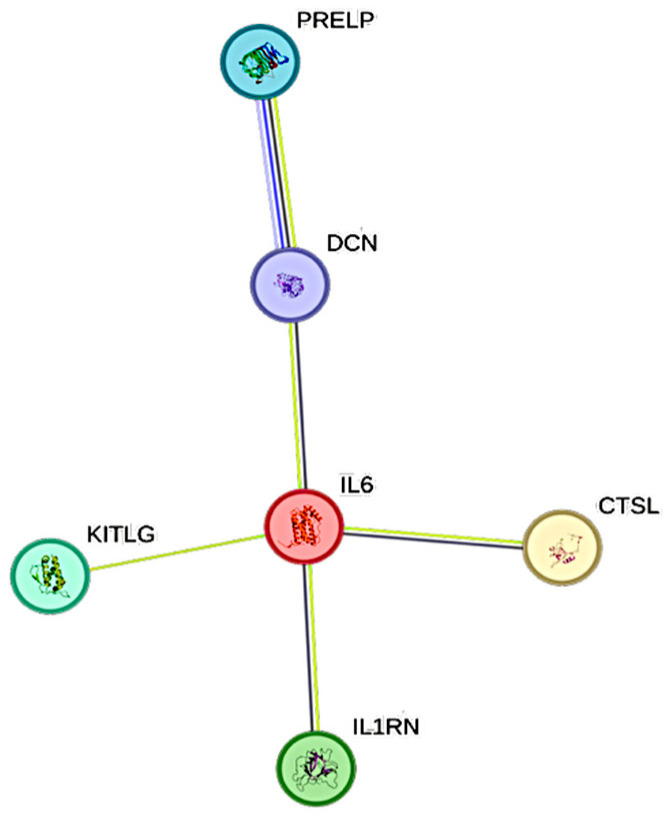
This figure represents a network of interactions among cytokines, demonstrating significant associations between circulating cytokine levels and CRP. The nodes (CCGFs) are connected by edges of varying thickness, indicating confidence of interactions.

**Figure 3 ijms-26-11419-f003:**
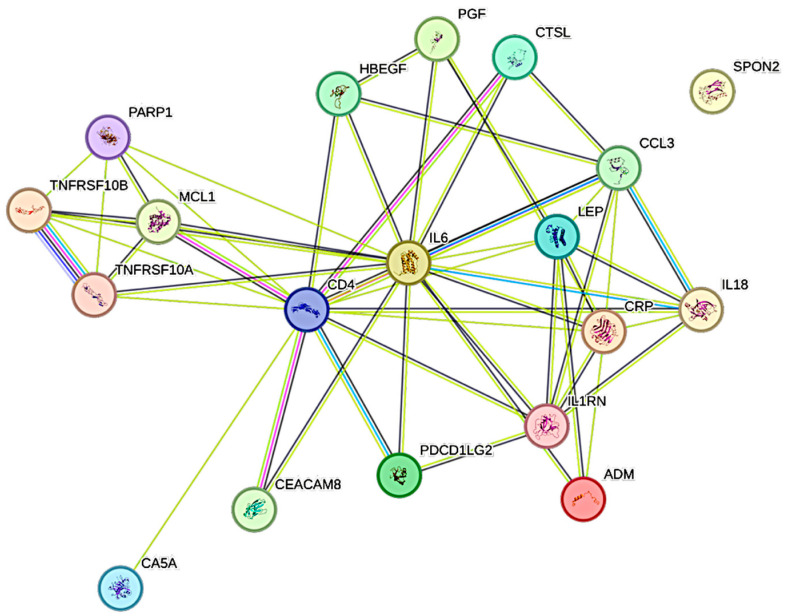
This figure represents a network of interactions among cytokines, demonstrating significant associations between circulating cytokine levels and procalcitonin. The nodes (CCGFs) are connected by edges of varying thickness, indicating confidence of interactions.

**Figure 4 ijms-26-11419-f004:**
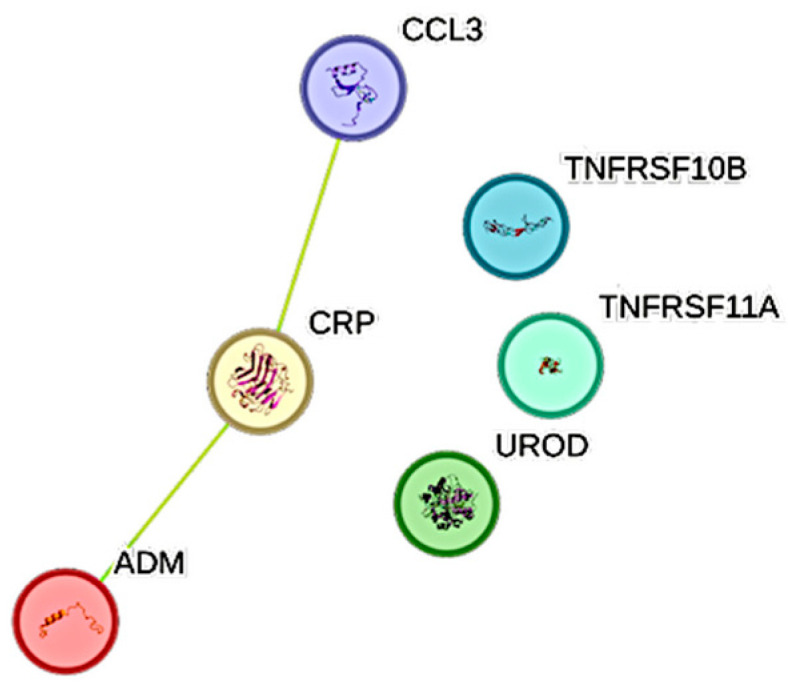
This figure represents a network of interactions among cytokines, demonstrating significant associations between circulating cytokine levels and IL6. The nodes (CCGFs) are connected by edges of varying thickness, indicating confidence of interactions.

**Figure 5 ijms-26-11419-f005:**
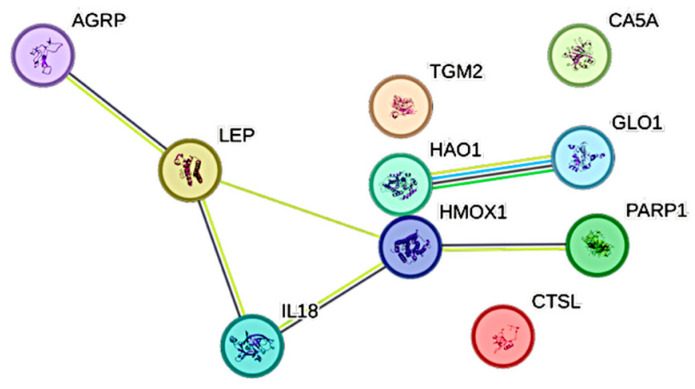
This figure represents a network of interactions among cytokines, demonstrating significant associations between circulating cytokine levels and ferritin. The nodes (CCGFs) are connected by edges of varying thickness, indicating confidence of interactions.

**Table 1 ijms-26-11419-t001:** Significant associations between biomarkers and cytokines, chemokines, and growth factors (CCGFs). Results were adjusted for multiple testing. Adjusted *p*-values below 0.10, corresponding to an expected false discovery rate of <10%, were considered statistically significant. Calculations of *p*-values are explained in Section 4.5.

Biomarker	CCGFs	*p*-Values	Benjamini–Hochberg Significance	Benjamini–Hochberg *p*-Value
Basophil	Gal9	0.001026	significant	0.02642037
Basophil	RAGE	0.000916	significant	0.02424253
Eosinophil	TNFRSF13B	0.000654	significant	0.01884023
Eosinophil	Gal9	0.004025	significant	0.05629887
Eosinophil	RAGE	0.003435	significant	0.05095177
Leucocyte	ADAMTS13	0.002680	significant	0.0444628
Leucocyte	GIF	0.001105	significant	0.02775048
Leucocyte	IL16	0.003129	significant	0.04925209
Leucocyte	CEACAM8	0.000155	significant	0.00583731
Lymphocyte	CTRC	0.004319	significant	0.05920063
Monocyte	THPO	0.007095	significant	0.08904599
Monocyte	CTSL1	0.000873	significant	0.02374733
Monocyte	TM	0.001577	significant	0.03375487
Neutrophil	ADAMTS13	0.001931	significant	0.03665511
CRP	IL6	0.000429	significant	0.01354379
CRP	CTSL1	0.006987	significant	0.08883657
CRP	IL1Ra	0.003948	significant	0.0560123
CRP	SCF	0.000631	significant	0.01871042
CRP	PRELP	0.008027	significant	0.09823623
CRP	DCN	0.004414	significant	0.05920063
Procalcitonin	ADM	0.000084	significant	0.00533579
Procalcitonin	CRP	0.000113	significant	0.00533579
Procalcitonin	IL6	0.002820	significant	0.04525296
Procalcitonin	PGF	0.000142	significant	0.00555488
Procalcitonin	PDL2	0.000224	significant	0.00804027
Procalcitonin	CTSL1	0.000052	significant	0.00422464
Procalcitonin	LEP	0.002546	significant	0.0444628
Procalcitonin	CA5A	0.001341	significant	0.03281978
Procalcitonin	CD4	0.000230	significant	0.00804027
Procalcitonin	PARP1	0.002607	significant	0.0444628
Procalcitonin	IL1ra	0.001637	significant	0.03409022
Procalcitonin	TNFRSF10A	0.000005	significant	0.00091348
Procalcitonin	TRAILR2	0.001493	significant	0.03375487
Procalcitonin	IL18	0.007520	significant	0.09318602
Procalcitonin	SPON2	0.005478	significant	0.07056722
Procalcitonin	TM	0.005266	significant	0.06966781
Procalcitonin	CEACAM8	0.000477	significant	0.01457828
Procalcitonin	CCL3	0.000257	significant	0.00867255
Procalcitonin	HBEGF	0.003396	significant	0.05095177
IL6	ADM	0.002004	significant	0.03702356
IL6	CRP	0.000429	significant	0.01354379
IL6	PCT	0.002820	significant	0.04525296
IL6	TNFRSF11A	0.005383	significant	0.07026409
IL6	TRAILR2	0.002533	significant	0.0444628
IL6	CCL3	0.001488	significant	0.03375487
Ferritin	CTSL1	0.000806	significant	0.02255561
Ferritin	TGM2	0.003699	significant	0.05325795
Ferritin	LEP	0.001493	significant	0.03375487
Ferritin	CA5A	0.003525	significant	0.05150071
Ferritin	PARP1	0.000019	significant	0.00209397
Ferritin	HAOX1	0.001544	significant	0.03375487
Ferritin	IL18	0.000031	significant	0.0030732
Ferritin	GLO1	0.004384	significant	0.05920063
Ferritin	HO1	0.000140	significant	0.00555488
Ferritin	AGRP	0.001586	significant	0.03375487

## Data Availability

The original contributions presented in this study are included in the article/Appendix A. Further inquiries can be directed to the corresponding author.

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
