# Peer review of "Mapping Interactions Between Cytokines, Chemokines, Growth Factors, and Conventional Biomarkers in COVID-19 ICU-Patients"

_ijms, 2025, doi:10.3390/ijms262311419_

Round 1

Reviewer 1 Report

Comments and Suggestions for Authors

The manuscript investigates associations between standard inflammatory biomarkers (CRP, ferritin, PCT) and circulating cytokines, chemokines, and growth factors (CCGFs) measured using the Olink PEA in critically ill COVID-19 patients admitted to the ICU within the prospective PronMed cohort. Although the dataset is valuable and the methodological framework is generally sound, several conceptual, statistical, and presentation-related issues must be addressed before the manuscript is suitable for publication. My detailed comments are outlined below.

Title

  • The current title is overly general and does not accurately reflect the study population or the specific Olink panel used. Consider clarifying that the study examines ICU patients and focuses on a cardiovascular-related cytokine panel.

Abstract

  • The abstract states that concentrations of 92 CCGFs were measured but that 114 patients were included. Please clarify the discrepancy between the number of analytes and the number of patients.
  • The novelty of the study is not clearly conveyed. Several associations reported here (e.g., CRP–IL-6, ferritin–IL-18, PCT–IL-6) are well established in prior COVID-19 literature. Please articulate more clearly what new insight this work provides compared to existing proteomic and cytokine studies.

Keywords

  • “Blood cells” and “Intensive care” are too broad. More specific, field-relevant terms are recommended.
  • “COVID” should be written as “COVID-19”.

Results

  • Figure 1: The heatmap lacks a clear legend indicating the meaning of the color scale and significance thresholds. A visual scale bar from blue to red with corresponding correlation values would improve interpretability.
  • Table 1: The table formatting should be improved and preferably fitted on a single page. The legend should explicitly state that significance was defined as p < 0.10.
  • Figure 2: The figure and its legend should appear on the same page.
  • Figure 3: The STRING network visualization is difficult to interpret. Labels are small, the layout is visually dense, and the biological relevance is unclear.
    STRING network figures need higher resolution and clearer legends describing colors, interaction scores, and thresholds. Consider simplifying or complementing them with alternative visualization approaches.
  • The statistical methodology, particularly the justification for using p < 0.10, requires further clarification.
  • Several figures and tables lack adequate legends or explanatory notes.
  • I recommend adding a brief heading or summary sentence for each results subsection to guide readers before presenting figures or detailed descriptions.

Discussion

  • The subsection “3.1” appears unnecessary and disrupts the flow. Removing it is recommended.
  • The discussion should more clearly address how this study advances current knowledge beyond previous COVID-19 cytokine and proteomic analyses.

Methods

  • Important clinical covariates that influence inflammatory markers (e.g., immunomodulatory treatments, bacterial co-infection, comorbidities, organ dysfunction) are not described. These should be reported or addressed as limitations.
  • Clarify sample handling procedures: plasma storage conditions, duration of storage, and number of freeze–thaw cycles.
  • “PATIENTS AND METHODS” should be changed to “Materials and Methods” in accordance with standard journal formatting.
  • Please justify briefly why the significance threshold was set at p < 0.10 rather than the conventional p < 0.05.

Conclusion

  • Consider adding a statement on the potential future applications or clinical relevance of the findings.

General Comments

  • Ensure consistent use of abbreviations throughout the manuscript (e.g., IL-6 vs IL6).
Comments on the Quality of English Language

The English language requires minor revision for clarity and precision.

Author Response

Reviewer 1.

The authors wish to thank the reviewer for valuable remarks on our manuscript, which will help us to further improve the presentation of our results and conclusions.

Our answers/changes of the manuscript are in red.

The manuscript investigates associations between standard inflammatory biomarkers (CRP, ferritin, PCT) and circulating cytokines, chemokines, and growth factors (CCGFs) measured using the Olink PEA in critically ill COVID-19 patients admitted to the ICU within the prospective PronMed cohort. Although the dataset is valuable and the methodological framework is generally sound, several conceptual, statistical, and presentation-related issues must be addressed before the manuscript is suitable for publication. My detailed comments are outlined below.

Title

  • The current title is overly general and does not accurately reflect the study population or the specific Olink panel used. Consider clarifying that the study examines ICU patients and focuses on a cardiovascular-related cytokine panel.
  • We agree with the Reviewer that the title needs improvement and suggest the following title:
    “Mapping Interactions Between Cytokines, Chemokines, Growth factors, and Conventional Biomarkers in COVID-19 ICU-patients”

Abstract

  • The abstract states that concentrations of 92 CCGFs were measured but that 114 patients were included. Please clarify the discrepancy between the number of analytes and the number of patients.
  • As stated in Section 4.3, The Olink Target 96 Cardiovascular II panel targets 92 proteins. 92 biomarkers were analyzed in each patient.
  • The novelty of the study is not clearly conveyed. Several associations reported here (e.g., CRP–IL-6, ferritin–IL-18, PCT–IL-6) are well established in prior COVID-19 literature. Please articulate more clearly what new insight this work provides compared to existing proteomic and cytokine studies.
  • It is true as the Reviewer says that several associations are already well-known in COVID-19. On the other hand, it might be misleading to exclude some associations just because they are already known. The Abstract has been shortened in order to focus to a lesser degree on the well-known associations pointed out by the Reviewer. The Abstract now reads:
  • Severe coronavirus disease 2019 (COVID-19) is characterized by systemic hyperinflammation with cytokine and chemokine release, alongside with elevations in conventional laboratory biomarkers such as C-reactive protein (CRP), ferritin, and procalcitonin (PCT). However, the interplay between cytokines, chemokines, growth factors (CCGFs), and standard biomarkers remains incompletely understood. Therefore, we aimed to evaluate associations between CCGFs and conventional biomarkers from a broad aspect, utilizing the prospective PronMed cohort of critically ill COVID-19 patients admitted to the intensive care unit (ICU) at Uppsala University Hospital. Plasma concentrations of 92 CCGFs were analyzed in each patient, using the Olink Target 96 Cardiovascular II panel and analyzed in relation to conventional biomarkers and peripheral blood cell counts. Associations were evaluated using Spearman rank correlations with Benjamini–Hochberg correction for multiple testing. A total of 114 patients (median age 61 years (IQR:19), 75% male, median SAPS-3 52 (IQR:10) were included. Significant correlations confirmed CRP as a robust surrogate of cytokine-driven inflammation. Ferritin was strongly associated with macrophage-related markers, including IL-18, sCD163-related factors, and PARP1. PCT correlated with a wide range of CCGFs, including ADM, PGF, TRAILR2, and IL-6. Blood cell subsets also showed distinct associations with CCGFs, suggesting functional connections between cytokine signaling and hematological disturbances. Our findings demonstrate that conventional biomarkers of inflammation in COVID-19 reflect complex and distinct interaction patterns with cytokines, chemokines, and growth factors. Mapping these associations improves understanding of COVID-19 immunopathology and may inform biomarker-guided risk stratification in critical illness.
  •  

Keywords

  • “Blood cells” and “Intensive care” are too broad. More specific, field-relevant terms are recommended.
  • “COVID” should be written as “COVID-19”.
  • The Reviewer is right and we have changed the keywords as follows: Keywords: Biomarker; Nucleated blood cells; COVID-19; Critical illness; CRP; Cytokine; Ferritin; Inflammation; Procalcitonin

Results

  • Figure 1: The heatmap lacks a clear legend indicating the meaning of the color scale and significance thresholds. A visual scale bar from blue to red with corresponding correlation values would improve interpretability.
  • It is true that a heatmap should be more or less self-explanatory. Thus, the following is added to the legend: “Warmer colors (red) indicate stronger positive correlations, whereas cooler colors (light blue to blue) indicate more expressed negative correlations. No color denotes absence of correlation.”
  • Table 1: The table formatting should be improved and preferably fitted on a single page. The legend should explicitly state that significance was defined as p < 0.10.
  • We agree with the Reviewer´s opinion that table 1 is very long, but we have not been able to adjust the width of the columns enough to have the table inserted “side-by-side” on a single page. Especially not, since we have to consider the margins that will be necessary during the printing process.

The following sentence is added to the legend: “Results were adjusted for multiple testing. Adjusted p-values below 0.10, corresponding to an expected false discovery rate of < 10%, were considered statistically significant.”

  • Figure 2: The figure and its legend should appear on the same page.
  • We agree that this change would make the figure and its legend both easier to follow and more aesthetic.
  • Figure 3: The STRING network visualization is difficult to interpret. Labels are small, the layout is visually dense, and the biological relevance is unclear.
    STRING network figures need higher resolution and clearer legends describing colors, interaction scores, and thresholds. Consider simplifying or complementing them with alternative visualization approaches.
  • The rationale for using STRING images is explained in Section 4.4. Besides, we have several times published such images in the IJMS
  • The statistical methodology, particularly the justification for using p < 0.10, requires further clarification.
  • Several figures and tables lack adequate legends or explanatory notes.
  • The legends to Figure 1 and Table 1 are updated.
  • I recommend adding a brief heading or summary sentence for each results subsection to guide readers before presenting figures or detailed descriptions.
  • We agree. A brief introductory heading is now placed just above Section 2.3.

Discussion

  • The subsection “3.1” appears unnecessary and disrupts the flow. Removing it is recommended.
  • This subsection is of limited value and we agree to remove it.
  • The discussion should more clearly address how this study advances current knowledge beyond previous COVID-19 cytokine and proteomic analyses.
  • The following is now added to the DISCUSSION section: Subsection “3.1” is now replaced with the following: “Our findings extend current knowledge by integrating cytokines, chemokines, growth factors, and standard biomarkers by revealing distinct and biologically coherent correlation patterns. This comprehensive mapping clarifies how conventional biomarkers reflect underlying immune pathways, which may contribute to more refined biomarker-guided risk stratification.”

Methods

  • Important clinical covariates that influence inflammatory markers (e.g., immunomodulatory treatments, bacterial co-infection, comorbidities, organ dysfunction) are not described. These should be reported or addressed as limitations.
  • In the RESULTS section, we have described the cohort: BMI, sex, age, hypertension, diabetes, median duration of COVID before ICU admission, SAPS-3, as well as CRP, ferritin, and PCT. Immunomodulatory treatment (dexamethason) is briefly described in METHODS.

Clinical covariates that influence inflammatory markers are not extensively described, since this was not a part of our study plan. This has been addressed as limitations in the DISCUSSION section.

  • Clarify sample handling procedures: plasma storage conditions, duration of storage, and number of freeze–thaw cycles.
  • In subsection 4.1, the following is now added: “Plasma samples were stored in -80°C until analyzed.”
  • “PATIENTS AND METHODS” should be changed to “Materials and Methods” in accordance with standard journal formatting.
  • This is now changed to MATERIALS AND METHODS
  • Please justify briefly why the significance threshold was set at p < 0.10 rather than the conventional p < 0.05.
  • Initially we used 0.05 as threshold for level of significance without adjustment for multivariate analysis. P-values were then sorted in increasing order (indicating that the 0.05 value does not affect further analysis) and used a false discovery rate at 10% as significant. This is an exploratory study and, in such studies, it is frequent to use somewhat higher levels of significance to ensure the avoidance of a statistical type II (beta) error. We have in several previous publications used 0.10 as the level of significance. This is now briefly explained in the “Statistical analysis” subsection.

Conclusion

  • Consider adding a statement on the potential future applications or clinical relevance of the findings.
  • We agree that such a statement would increase the value of this publication. Hence, the following is inserted in the CONCLUSIONS section. “These results may turn out to have clinical implications by clarifying which routinely measured biomarkers that best reflect specific immune pathways, improve risk stratification, and guide monitoring strategies in critical care.”

General Comments

  • Ensure consistent use of abbreviations throughout the manuscript (e.g., IL-6 vs IL6).
  • The abbreviations are revised.

The English language requires minor revision for clarity and precision.

An independent linguistic revision is performed.

Reviewer 2 Report

Comments and Suggestions for Authors

This study provides an integrated view of the inflammatory landscape in severe COVID-19 by linking routine biomarkers with a broad panel of cytokines, chemokines, and growth factors, indicating that common laboratory markers remain informative substitutes of complex immune activity and offer meaningful insight for monitoring and risk management in critically ill patients. However, there are several issues could be addressed before the manuscript can be considered for publication.

  1. Linking the observed biomarker–cytokine patterns to clinically relevant outcomes would help translate these findings into actionable insights and substantially enhance the study’s impact.
  2. It would be helpful to adjust the biomarker–CCGF associations for potential confounding factors including age, sex, BMI, comorbidities, and dexamethasone treatment. Doing so would clarify whether these correlations truly reflect COVID-19–related inflammation rather than underlying patient characteristics.
  3. A brief comparison with biomarker–cytokine patterns reported in other viral pneumonias or non-COVID sepsis cohorts would help clarify whether the identified pathways are unique to COVID-19 or reflect broader features of critical illness.

Author Response

Reviewer 2.

Dear Reviewer! We appreciate your relevant comments and constructive criticism. Our answers are in red.

This study provides an integrated view of the inflammatory landscape in severe COVID-19 by linking routine biomarkers with a broad panel of cytokines, chemokines, and growth factors, indicating that common laboratory markers remain informative substitutes of complex immune activity and offer meaningful insight for monitoring and risk management in critically ill patients. However, there are several issues could be addressed before the manuscript can be considered for publication.

  1. Linking the observed biomarker–cytokine patterns to clinically relevant outcomes would help translate these findings into actionable insights and substantially enhance the study’s impact.
  2. It would be helpful to adjust the biomarker–CCGF associations for potential confounding factors including age, sex, BMI, comorbidities, and dexamethasone treatment. Doing so would clarify whether these correlations truly reflect COVID-19–related inflammation rather than underlying patient characteristics.
  3. A brief comparison with biomarker–cytokine patterns reported in other viral pneumonias or non-COVID sepsis cohorts would help clarify whether the identified pathways are unique to COVID-19 or reflect broader features of critical illness

We find all these remarks adequate and highly valuable and we think that our responses can be summarized as follows:

Although the sampling in the PronMed cohort was performed prospectively, this analysis was performed retrospectively. Thus, the associations that we noted between inflammatory biomarkers and CCGFs have not had any impact on the management. The Olink panel used was designed for measuring cardiovascular proteins, not inflammatory ones, suggesting that dexamethasone would have a limited impact on our results. From a previous study of ours (PMID: 40869066), we know that even in assumedly healthy people, IL6 is associated with body mass index, high density lipoproteins, estimated glomerular filtration rate, and triglycerides but not with sex or age. The question whether the identified pathways are unique to COVID-19 or reflect critical illness more generally follows logically from the two initial questions and may be hence be seen as a unit, which we would like to answer by adding the following into the DISCUSSION section:

“In this exploratory trial, we used a cardiovascular Olink panel to evaluate a broad spectrum of CCGFs reflecting cardiac injury in patients infected with SARS-CoV-2. It is a drawback that some important clinical covariates that influence inflammatory markers (e.g., immunomodulatory treatments, bacterial co-infection, comorbidities, organ dysfunction) are not evaluated regarding their potential impact on the CCGFs measured. As expected, elevated levels of IL6, CRP, ferritin, and PCT were observed, all of which have been associated with cardiovascular events (PMID: 32344321; PMID: 32719447; PMID: 33228225). Treatment with dexamethasone in severe COVID-19 may have attenuated some of the CCGFs analyzed. In a previous study, we demonstrated that IL6 was significantly associated with body mass index, high-density lipoproteins, estimated glomerular filtration rate, and triglycerides in presumed healthy individuals (PMID: 40869066). Thus, it cannot be excluded that certain comorbidities may influence the interpretation of IL6 levels, and possibly those of other CCGFs associated with conventional biomarkers and nucleated blood cells. Viral pneumonia has been predicted by IL6, IL27, and CRP, with distinct cytokine expression profiles differentiating viral from bacterial community-acquired pneumonia (PMID: 31310442). In SARS-CoV-2 infection, IL6 is one of the cytokines most clearly associated with poor clinical outcomes (PMID: 33153908). Thus, prospective validation studies—such as cluster analyses using artificial intelligence—may help determine whether specific cytokine patterns can guide therapeutic decision-making. If so, the Olink panel (or similar platforms) may be cost-effective, particularly when evaluated in relation to ICU care costs.”

Round 2

Reviewer 1 Report

Comments and Suggestions for Authors

All previously raised comments have been adequately addressed by the authors. The revised manuscript shows clear improvement in formatting, clarity, and scientific accuracy. I am satisfied with the revisions and have no further concerns.

I recommend the manuscript for publication in its current form.

Reviewer 2 Report

Comments and Suggestions for Authors

The manuscript has improved following the authors’ revisions. The previous issues regarding clarity and contextual framing have been adequately addressed. With these improvements, the overall presentation now meets the standard for publication.